# Utility analysis of digital villages to empower balanced urban-rural development based on the three-stage DEA-Malmquist model

**Lingling Cao**[1,2], **Huawei Niu**[1]*, **YiFeng Wang**[2]

**1** School of Economics and Management, China University of Mining and Technology, Xuzhou, Jiangsu, China, **2** Business School, Suqian University, Suqian, Jiangsu, China

* lb21070039@cumt.edu.cn

## Abstract

Rural subjects, the agricultural industrial structure, public services and rural governance are fully empowered by digital villages. This empowerment effectively compensates for the urban-rural digital divide and promotes the equalization of urban-rural income, consumption, education, medical care, and governance. Based on the three-stage data envelopment analysis (DEA) model and Malmquist index, this article conducts an in-depth study of the static and dynamic efficiency trends of digital villages that empower urban-rural balanced development in 31 provinces in China from 2015 to 2020. The results show that comprehensive technical efficiency of 31 provinces is weak DEA effective, and that the scale efficiency is the main factor affecting comprehensive technical efficiency. The educational level, local finance and industrial structure optimization have a significant positive impact on efficiency evaluation, but technological innovation and the urbanization level have a significant negative impact. Total factor productivity shows diminishing marginal utility based on the Malmquist index and its decomposition change. Restricted by the change in technological progress, the efficiency of digital villages in China in enabling urban-rural equilibrium needs to be further improved.

## 1 Introduction

Since the digital village development strategy was first proposed in the No.1 Central Document in 2018, favorable policies have frequently appeared. The benefits and knowledge spillover effects of digital technologies on agriculture, rural areas and farmers should be fully exploited to facilitate rural revitalization. At the beginning of 2021, the Central Rural Conference clearly proposed that digital technology empowerment is an important focus and inevitable choice for realizing rural revitalization and promoting urban-rural balanced development.

"Digital divide" is defined as the disparity between 'the information-rich' and 'the information-poor', especially in access to new information technology such as personal computers (PC) and the internet [1]. Currently, a mobile phone is an important social communication tool and a multifunctional medium. Thus, inequality with regard to possession and use of

2015-2020 National Statistical Yearbook, Provincial and Municipal Statistical Yearbooks, China Urban Construction Statistical Yearbook, China internet Development Report, and National County Digital Agriculture and Rural Development Level Evaluation Report. All variables are processed in a positive direction, and input–output variables meet a significant correlation.

**Funding:** This research was supported by the Fundamental Research Funds for the Central Universities (2022ZDPYSK02, HN), the National Natural Science Foundation of China (71871120, HN), the Excellent Social Science Application Engineering Projects of Jiangsu Province (21SYB-091, LC), and the "Blue Project" of Jiangsu Universities. The funders had no role in study design, data collection and analysis, decision to publish, or preparation of the manuscript.

**Competing interests:** The authors have declared that no competing interests exist.

mobile media creates gaps among different groups, known as the 'Mobile divide' [2]. The "mobile divide" and "digital divide" have been parallel phenomena and have severely hindered the balanced and healthy development of urban-rural areas. In the digital age, digital rural strategy has become a new topic for the construction of urban-rural relationships, and an inclusive digital urban-rural integration mechanism has become the best path for mutual promoting and interconnecting urban and rural areas [3].

A consensus has been reached regarding the full and efficient development of rural social resource endowments with digital technology, bridging the urban-rural digital and economic gap and resolving the contradiction between the dual economic structures. It is extremely important to explore the mechanism and the effect of digital rural strategy that empowers the balanced development of urban and rural areas. Many studies have been conducted based on digital village empowerment.

First, digital empowerment is the extension and development of empowerment theory in the information age, and it is a new theory stemming from the continuous maturation of digital technology. Digital rural empowerment refers to the use of new digital technologies to continuously improve the endogenous power of the rural economy, rural culture and rural governance [4]. Digital technology empowers rural revitalization to face unprecedented opportunities and challenges. (1) The structural reform of the agricultural supply side is deepening, the policy context for digital technology to empower rural revitalization and development is becoming stronger, the economic and material foundations are more solid, and innovation-driven demand is becoming stronger [5]. (2) The top-level design of digital technology empowering rural revitalization and high-quality agricultural development urgently needs to be perfected, but the lack of talent and the relative lag in "new infrastructure" have become real problems. Digital technology has become an important constraint on enabling high-quality agricultural development [6].

Second, rural entities use digital tools to empower endogenous elements. Such tools have been comprehensively improved for survival, management, operation and technical capabilities. They are conducive to promoting the transformation and upgrading of traditional agricultural management models [3, 6]. Digital technology is embedded in the system of agricultural factor allocation, industry, production, operation and circulation. It accurately locates the entire agricultural industry chain, promotes the green allocation of agricultural resources, deepens the integration of industries, promotes intelligent production management and smart agriculture-related circulation, and help to completely eliminate the digital divide [7–9]. Digital technology has resulted in new value attributes, such as the integration of rural governance resources and multidimensional interactions. It effectively promotes digital, intelligent and interconnected development and shifts the traditional rural governance paradigm [10, 11].

Third, the integration of digital technology and agricultural economic development can help improve rural economic development and the knowledge and capabilities of farmers, and it can promote agricultural upgrading, rural progress and farmer development in a comprehensive manner [12]. Taobao Village, which have undoubtedly brought economic prosperity to rural areas, are regarded by both academia and the government as an effective means of revitalizing rural areas and narrowing the rural–urban gap [13–15]. The scale and accumulation effect of the "digital economy plus industrial layout and supply chain extension" are conducive to increasing the economic added value of agricultural products and bridging the potential economic gap between urban and rural imbalances [6, 16]. Regarding the implementation of urban-rural policies, digital technology drives the dilution of regional characteristics. In addition, the borderless information collection ability promotes the integration of urban and rural areas [17]. A new dual-center policy system of "rural characteristic elements plus urban advantage elements" has been formed [18]. This system will help prevent the "migration

boom" and promote the return of resources, technology, and talent to agricultural and rural areas. Digital technology injects new momentum into the development of cultural interconnection between urban and rural areas, helps to fully realize the barrier-free sharing of urban and rural culture, and realizes the parallel complementation and integration of multiple urban and rural cultures [19]. For rural governance, digital technology is used as a driving force to empower the field of rural governance, stimulate villagers' initiative and build a team of professional governance talent [20]. Doing so requires the integration of shared data resources to realize an "online plus offline, technology and system" new digital rural public service system across regions and departments [21]. For example, the emergence of e-commerce associations in Taobao Villages suggests a larger but contained space for rural e-tailers' participation in public affairs, leading to a new party-state corporatist mode of rural governance [14].

Most studies emphasize that digital technology is a practical need for urban-rural balanced development, but they seldom pay attention to the path of balanced urban and rural development based on digital rural empowerment. Research at this stage is based more on the theoretical level to build a logical framework for digital villages and enable the high-quality development of agriculture, and a quantitative utility evaluation of digital villages is lacking. This article comprehensively and systematically identifies the transmission mechanism of digital villages to enable urban-rural balanced development, and it actually measures the development of digital villages in different locations of China, based on the three-stage data envelopment analysis (DEA) model. This will provide intellectual support for the comprehensive realization of digital village construction.

## 2 Mechanism of action under the input–output framework

Based on the input-output framework, we comprehensively identified various transmission channels for digital villages to enable balanced urban and rural development (Fig 1).

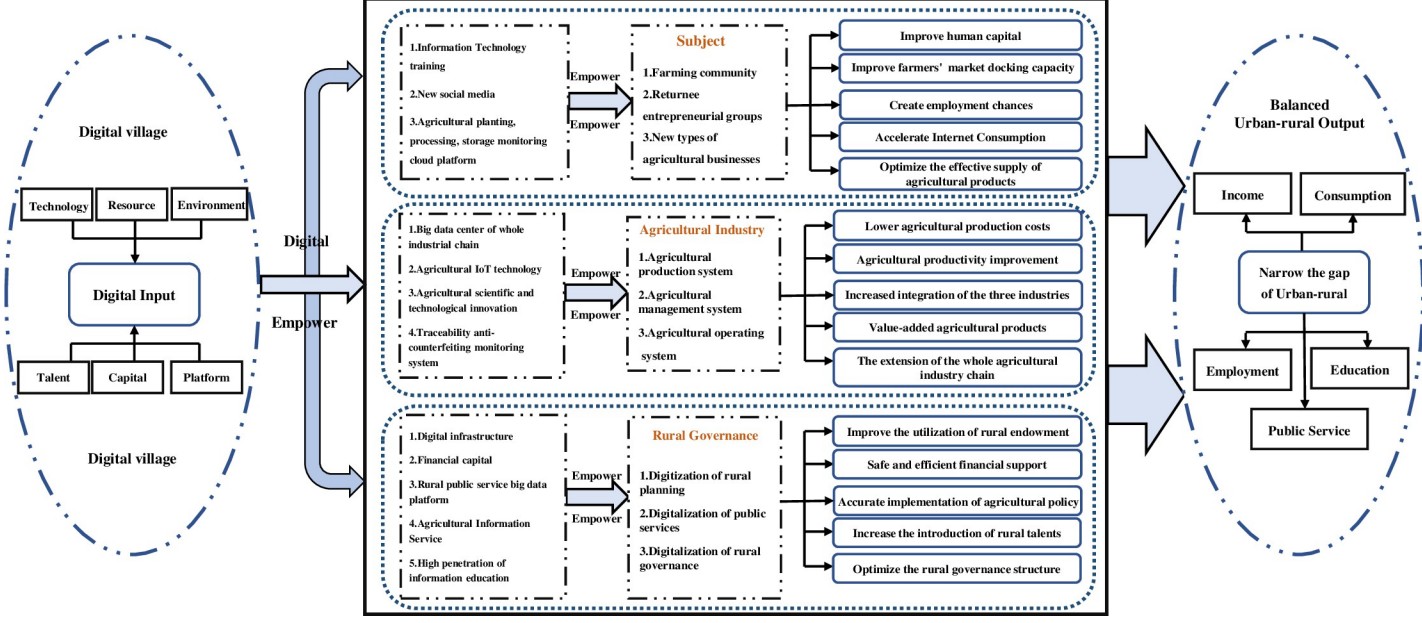

**Fig 1. The mechanism of enabling urban-rural equilibrium.**

## 2.1 Empowering agricultural subjects

**2.1.1 Digital education empowers digital literacy.** Human capital theory confirms that digital technology training for rural subjects can effectively stimulate farmers' ability and willingness to use digital technology and strengthen the technical thinking and modernization consciousness of rural subjects [22]. Comprehensive coverage of the whole process and comprehensive digital education should be carried out for rural subjects to promote the enthusiasm and initiative of those who rely on agriculture, and those who return to their hometowns to start businesses and new agricultural operation projects. The popularization of digital education comprehensively inspires the digital universe, and the social, creative and safety literacy of rural subjects. The digital application of rural subjects continues to extend vertically and horizontally, thus driving the continuous growth in and quality of rural internet consumption and narrowing the gap in urban-rural consumption. Multiple channels should be built to fully cover the learning needs of rural subjects and to promote the breadth and depth of rural subjects' direct participation in digital life.

**2.1.2 Digital platforms enable the utilization of digital technologies.** At the production end of farming and returning entrepreneurial groups, farmers use new social media platforms with a low threshold to promote agricultural and sideline products through multiple channels and accurately adjust sales strategies based on market feedback so that they can improve their ability to connect with the market, expand sales and increase income. Improving the human capital of typical rural subjects has the spillover effect and knowledge diffusion effect of driving the demonstration effect [4]. One person drives a group of people, and one region drives the surrounding region or even infinitely replicates beyond the regional boundary, which has a great driving effect on rural employment.

For new agricultural operators, building a monitoring cloud platform system can help accurately predict market demand and improve quality and efficiency [23]. A digital platform for the circulation of agricultural and sideline products organically connects farmers, middlemen, dealers and other entities to fully share data and information, which is conducive to reducing operating costs. Agglomeration using platforms helps realize information, and resources. Moreover, the information asymmetry of barriers will be broken; thus, using its scale advantage, the platform will constantly attract market participation, the realization of agricultural products and the efficient matching of market supply and demand and increase added value of agriculture.

**2.1.3 Digital infrastructure empowers digital behavior.** The construction of rural digital infrastructure is conducive to realizing the accessibility and equality of rural subjects' information, blocking the intergenerational transmission of the lack of information ability among rural subjects, and comprehensively stimulating potential digital demand and data acquisition ability [8]. Digital education can empower rural subjects with digital literacy and guide rural subjects in an orderly manner to consume internet finance, online travel and other fields of deep internet application.

## 2.2 Empowering the agricultural industry

**2.2.1 Empowering the precision of the agricultural production system.** Digital technology empowers agricultural production systems to achieve the integration of agriculture and other industries. First, it promotes the application of the agriculture and forestry "four situations" monitoring system, unmanned aerial vehicle (UAV) flight defense, intelligent irrigation and fertilization, intelligent greenhouse construction, precision feeding and other agricultural production fields to promote the modernization and precision of agricultural production, minimize agricultural production costs and effectively avoid agricultural business risks [24].

Second, by virtue of digital technology, products are a high-end, green and standardized brands, maximizing exports to the international market and improving the current situation of China's agricultural trade deficit [25]. Third, it is necessary to accelerate the return and concentration of urban related agricultural enterprises and accelerate the intensive and large-scale operation of agriculture by relying on smart agricultural science and technology industrial parks.

**2.2.2 Empowering an efficient agricultural operation and management system.** Digital technology promotes the precise and optimized allocation of agricultural production factors, complements the short board of the agricultural industry chain, and effectively promotes the establishment of a relationship between the production end and the consumption end [25]. Relying on an agricultural information service (AIS) can boost the brand promotion of agricultural products, integrate agricultural experience, handicrafts, leisure tourism and other elements, and invigorate the interest of all parties in the agricultural industry chain. Building a digital information decision-making platform system is necessary to promote the rapid decision-making response of agricultural operation subjects, improve the accuracy of decision-making, and realize the efficient operation of agricultural operation systems.

### 2.3 Empowering rural governance

**2.3.1 Empowering the digitalization of rural planning.** Ecological planting and breeding zones with distinctive regional endowments should be built based on local conditions, and the layout of the village road network should be optimized to eliminate dead-end roads and achieve uniform transportation between urban and rural areas. Relying on a big data platform will help build a new type of livable rural community with an appropriate scale and complete functions, and it will enhance the awareness and provision of social services to rural households. In rural planning and construction, cement irrigation systems should be avoided to prevent the destruction of the original ecological food chain and avoid backtracking and resource waste.

**2.3.2 Enabling public services to be digitized.** In basic compulsory education, the construction of rural digital campuses should be vigorously promoted. Digital education infrastructure and cloud platforms for the sharing of educational resources should be built to meet the hardware infrastructure requirements of rural digital compulsory education. In mass education, digital technology should be adopted to build developmental digital infrastructure, such as urban-rural interconnections, digital TV, and digital libraries. The multichannel construction of learning resources is conducive to the realization of educational equality.

By leveraging new digital technologies, an integrated smart medical platform for urban and rural areas should be built to form a comprehensive medical network system integrating expert databases, and patient information and medical records, and to achieve seamless connection between high-quality medical resources in cities and rural areas [26]. A big data platform promotes the interconnection and data sharing between rural medical and health institutions and urban hospitals and realizes the online settlement of medical insurance in different places.

New digital technologies will be used to build digital application platforms for rural life, such as smart rural logistics systems, smart monitoring systems and network interaction systems, to form a complete, closed-loop service system for rural people. With the help of AISs, policies and regulations on agricultural subsidies, the publicity of village affairs, agricultural production and sales markets can be made public and transparent.

**2.3.3 Enabling the digitalization of rural governance.** The level of digitalization can significantly promote the accurate identification of low-income rural groups, eliminate information asymmetry within rural areas, and improve the accurate identification rate. The layer-by-

layer implementation of the top-level system needs to fully rely on blockchain technology and rural information public service platforms. Only in this way, can we improve the transparency of financial support for agriculture and the precision of policy implementation.

The capacity gap and information gap between local governments and rural participants are gradually narrowing [11]. The consciousness of rural subjects to participate in rural governance is gradually enhanced, which fundamentally improves the participation of local people in rural governance. Digital technology has broken the traditional spatial pattern of governance, and rural governance entities have realized cross-regional and cross-temporal deep communication and interaction.

A data cloud service platform provides cloud services for application assistance, medical subsidies and other businesses, and it improves the efficiency of public services. Smart government departments can accurately identify the public demands of rural subjects and realize "proactive plus precise" service-oriented rural governance. They can carefully consider the hidden dangers existing in various subjects in rural areas in a timely manner and realize the governance mode of "tracking after the event" and "warning in advance".

## 3 Method

The DEA model was first proposed by Charnes and Cooper in 1978 to calculate the relative efficiency of multiple inputs and multiple outputs in decision-making units (DMUs). Fried (2002) innovatively proposed a three-stage DEA-Malmquist model, which avoids the disadvantages of traditional DEA models, such as management inefficiency, environmental factors, and random interference.

$$min\theta_s - \varepsilon(\sum_{i=1}^{m} S_{is}^- + \sum_{j=1}^{n} S_{js}^+)$$

$$s.t. \begin{cases} \sum_{s=1}^{K} \lambda_s X_{is} + S_{is}^- = \theta_s X_{is}, i = 1, 2, \cdots, m \\ \sum_{s=1}^{k} d\lambda_s Y_{js} - S_{js}^+ = Y_{js}, j = 1, 2, \cdots, n \\ \lambda_s \geq 0, \sum_{s=1}^{k} \lambda_s = 1, s = 1, 2, \cdots, K \\ S_{is}^- \geq 0, S_{js}^+ \geq 0 \end{cases} \quad (1)$$

where $\theta_s$ is the overall efficiency value, $X_{is}$ is the input indicator, $Y_{js}$ is the output indicator, $S_{is}^-$ and $S_{js}^+$ represent the input slack variable and the output slack variable, respectively, $\lambda_s$ is the weight variable, $\varepsilon$ is infinitesimal, and $d$ is the progressive movement factor of the system. The input redundancy value can be calculated by the following formula:

$$\triangle X_{is} = X_{is} - \sum_{s=1}^{k} \lambda_s X_{is} = (1 - \theta_s)X_{is} + S_{is}^-, i = 1, 2, \cdots, m \quad (2)$$

In the second stage, the panel stochastic frontier analysis (SFA) model is used to measure and adjust the value of input redundant variables. Taking the input redundancy value $\triangle X_{is}$ as the independent variable and external environmental factors as the independent variable, the panel SFA model is constructed to further eliminate invalid data in the slack variable input in

the first stage:

$$\Delta X_{is} = f^i(Z_{st}; \beta^i) + v_{ist} + u_{ist}, i = 1, 2, \cdots, m; s = 1, 2, \cdots, k; t = 1, 2, \cdots, T \tag{3}$$

Where $f^i(Z_{si}; \beta^i)$ is the definite influence of the external environment on input redundancy; $Z_{st} = [Z_{1t}, Z_{2t}, \cdots, Z_{Ht}]$ is the environment variable vector; $H$ is the number of environment variables; $\beta^i$ is the environmental variable estimation parameter; $v_{ist}$ and $u_{ist}$ are random interference items and management invalid noise, respectively, $v_{ist} \sim N(0, \sigma_{iv}^2)$. $u_{ist}$ is the normal distribution truncated at zero $u_{ist} \sim N^+(0, \sigma_{iu}^2)$.

The adjusted results are as follows:

$$\hat{X}_{ist} = X_{ist} + [max(Z_{st}\hat{\beta}^i) - Z_{st}\hat{\beta}^i] + [max(\hat{v}_{ist}) - v_{ist}]$$
$$i = 1, 2, \cdots, m; s = 1, 2, \cdots, k; t = 1, 2, \cdots, T \tag{4}$$

where $X_{ist}$ is the initial investment index; $\hat{X}_{ist}$ is the adjusted input variable value; $[max(Z_{st}\hat{\beta}^i) - Z_{st}\hat{\beta}^i]$ indicates adjustments to the external environment; $[max(\hat{v}_{ist}) - v_{ist}]$ is used to increase all DMUs to the same level of luck. To obtain the random error term, it is necessary to separate management invalid noise $u_{ist}$ from mixed error term $\varepsilon_{ist} = v_{ist} + u_{ist}$. The separation formula is as follows:

$$E(u_{ist}|v_{ist} + u_{ist}) = \frac{\lambda\sigma}{1 + \lambda^2}[\frac{\varphi(\varepsilon_{ist}\lambda/\sigma)}{\Phi(\varepsilon_{ist}\lambda/\sigma)} + \frac{\varepsilon_{ist}}{\sigma}] \tag{5}$$

Where $\sigma^2 = \sigma_v^2 + \sigma_u^2$, $\gamma = \frac{\sigma_u^2}{\sigma_v^2 + \sigma_u^2}$, $\lambda = \frac{\sigma_{iu}}{\sigma_{iv}}$, and $\varphi$ and $\Phi$ represent the density function and distribution function of the standard normal distribution, respectively. The final random error term is as follows:

$$E(v_{ist}|v_{ist} + u_{ist}) = \hat{X}_{ist} - Z_{st}\hat{\beta}^i - E(u_{ist}|v_{ist} + u_{ist}) \tag{6}$$

In the third stage, based on Formula (4), the adjusted input variable and the original output variable repeat the first-stage operation to obtain adjusted comprehensive efficiency value $\theta_s'$. Furthermore, based on the difference between the optimal investment $X_{ist}'' = \theta_s' X_{ist} - S_{ist}^-$ obtained by SFA and actual investment $X_{ist}$, investment redundancy value $\Delta X_{ist} = X_{ist} - X_{ist}'' = (1 - \theta_s')X_{ist} + S_{ist}^-$ is obtained.

The Banker–Charnes–Cooper (BCC)-DEA model cannot be used to analyze the dynamic technical efficiency changes of the DMU, but the Malmquist index can effectively solve this problem. Ray and Desli (1997) innovatively proposed the RD model of Malmquist exponential decomposition. The decomposition formula is as follows:

$$M_{RD}(X^t, Y^t, X^{t+1}, Y^{t+1}) = \frac{D_V^{t+1}(X^{t+1}, Y^{t+1})}{D_V^t(X^t, Y^t)} \times \left[\frac{D_V^t(X^t, Y^t)}{D_V^{t+1}(X^t, Y^t)} \times \frac{D_V^{t+1}(X^t, Y^t)}{D_V^{t+1}(X^{t+1}, Y^{t+1})}\right]^{\frac{1}{2}}$$
$$\times [\frac{D_C^t(X^{t+1}, Y^{t+1})/D_V^t(X^{t+1}, Y^{t+1})}{D_C^t(X^t, Y^t)/D_V^t(X^t, Y^t)} \times \frac{D_C^{t+1}(X^{t+1}, Y^{t+1})/D_V^{t+1}(X^{t+1}, Y^{t+1})}{D_C^{t+1}(X^t, Y^t)/D_V^{t+1}(X^t, Y^t)}]^{\frac{1}{2}} \tag{7}$$

where $X_s^t = (X_{1st}, X_{2st}, \cdots, X_{mst})^T$ is the investment index and $Y_s^t = (Y_{1st}, Y_{2st}, \cdots, Y_{nst})^T$ is the output indicator. When the return to scale is variable, the distance function of $(X_s^t, Y_s^t)$ in the t-th period is $D_V^t(X^t, Y^t)$; when the return to scale is constant, the distance function of $(X_s^t, Y_s^t)$ in the t-th period is $D_C^t(X^t, Y^t)$.

## 4 Measurement results

### 4.1 Index system and data selection

**4.1.1 Index system.** The input dimensions are based on three aspects: digital infrastructure, capital and data platform investment. Table 1 shows that the output dimensions based on five perspectives: equilibrium in the urban-rural economy, employment, consumption, social security and digital living environment [27]. The environmental change in urban-rural equilibrium efficiency is mainly affected by factors such as the economic environment, local finance, the industrial structure, technological innovation, and educational level. They provide a material foundation, the efficient matching of resources, industrial structure upgrading, sources of innovation, and intellectual support.

**4.1.2 Data sources.** For the variables, province-level data from 2015 to 2020 are selected. The data sources are the 2015–2020 National Statistical Yearbook, Provincial and Municipal Statistical Yearbooks, the China Urban Construction Statistical Yearbook, the China Internet Development Report, and the National County Digital Agriculture and Rural Development Level Evaluation Report. All variables are processed in a positive direction, and input–output variables have a significant correlation.

### 4.2 Result analysis

**4.2.1 The first stage.** We select original data from 2015 to 2020 and use MAXDEA software to calculate the comprehensive technical efficiency, pure technical efficiency, scale efficiency and investment target values of digital rural villages to empower urban-rural balanced development in 31 provinces (Table 2). The input slack variable is calculated according to the input original value and input target value. The three types of efficiencies that enable the balanced development of urban and rural areas in digital villages across the country all show a downward trend, indicating that the marginal efficiency of input is diminishing. The national comprehensive technical efficiency, pure technical efficiency and scale efficiency average are

**Table 1. Index system.**

| | Level indicators | Secondary indicators | Symbol | Attributes |
|---|---|---|---|---|
| Input | Digital infrastructure | Mobile internet penetration | $I_1$ | Positive |
| | | Fiber optic cable length per square kilometer per person | $I_2$ | Positive |
| | Capital investment | Per capita investment in fixed assets of the telecommunication industry | $I_3$ | Positive |
| | | Rural per capita investment in the construction of municipal public facilities | $I_4$ | Positive |
| | Digital platform | Cumulative count of AISs | $I_5$ | Positive |
| | | Number of rural public opinion monitoring platforms | $I_6$ | Positive |
| Output | Urban-rural economy | Ratio of urban and rural per capita disposable income | $O_1$ | Inverse |
| | Urban-rural employment | Proportion of the employed population in the secondary and tertiary industries/the primary industry | $O_2$ | Positive |
| | Urban-rural consumption | Ratio of the per capita consumption expenditure of urban and rural households | $O_3$ | Inverse |
| | Urban-rural social security | Ratio of the urban and rural endowment insurance participation rates | $O_4$ | Inverse |
| | Urban-rural digital living environment | Number of digital devices per 100 households in rural areas at the end of the year | $O_5$ | Positive |
| Environment | Economic | GDP per capita | $E_1$ | Positive |
| | | Urbanization rate | $E_2$ | Positive |
| | Finances | Revenue | $E_3$ | Positive |
| | Industrial structure | Proportion of the tertiary industry | $E_4$ | Positive |
| | Technological innovation | Number of granted invention patents | $E_5$ | Positive |
| | Educational level | Teacher-student ratio in compulsory education | $E_6$ | Positive |

**Table 2. First-stage and third-stage efficiency values.**

| Province | First stage | | | | Third stage | | | | Floating ranking | Efficiency increase |
|---|---|---|---|---|---|---|---|---|---|---|
| | Comprehensive technical efficiency | Pure technical efficiency | Scale efficiency | Rank | Comprehensive technical efficiency | Pure technical efficiency | Scale efficiency | Rank | | |
| Beijing | 0.8775 | 1.0000 | 0.8775 | 20 | 0.9961 | 1.0000 | 0.9961 | 12 | -8 | 0.1352 |
| Tianjin | 0.9322 | 1.0000 | 0.9322 | 7 | 0.9807 | 1.0000 | 0.9807 | 22 | 15 | 0.0520 |
| Hebei | 0.9180 | 0.9838 | 0.9334 | 13 | 0.9904 | 0.9973 | 0.9931 | 17 | 4 | 0.0789 |
| Shanxi | 0.9507 | 0.9961 | 0.9540 | 4 | 0.9965 | 0.9985 | 0.9980 | 11 | 7 | 0.0482 |
| Inner Mongolia | 0.9069 | 0.9946 | 0.9116 | 15 | 0.9926 | 0.9993 | 0.9933 | 15 | 0 | 0.0945 |
| Liaoning | 0.9428 | 1.0000 | 0.9428 | 5 | 1.0000 | 1.0000 | 1.0000 | 1 | -4 | 0.0607 |
| Jilin | 0.8504 | 0.9478 | 0.8947 | 24 | 0.9494 | 0.9683 | 0.9804 | 24 | 0 | 0.1164 |
| Heilongjiang | 0.9235 | 1.0000 | 0.9235 | 9 | 0.9986 | 0.9989 | 0.9997 | 8 | -1 | 0.0813 |
| Shanghai | 0.8618 | 0.9932 | 0.8668 | 22 | 0.9986 | 0.9998 | 0.9988 | 9 | -13 | 0.1587 |
| Jiangsu | 0.6812 | 0.8759 | 0.7770 | 29 | 0.9376 | 0.9710 | 0.9654 | 26 | -3 | 0.3764 |
| Zhejiang | 0.6209 | 1.0000 | 0.6209 | 31 | 0.9723 | 1.0000 | 0.9723 | 23 | -8 | 0.5660 |
| Anhui | 0.9212 | 0.9940 | 0.9269 | 11 | 1.0000 | 1.0000 | 1.0000 | 1 | -10 | 0.0855 |
| Fujian | 0.6734 | 0.7947 | 0.8480 | 30 | 0.8610 | 0.9025 | 0.9528 | 31 | 1 | 0.2786 |
| Jiangxi | 0.9217 | 0.9992 | 0.9224 | 10 | 0.9922 | 0.9996 | 0.9926 | 16 | 6 | 0.0765 |
| Shandong | 0.7752 | 0.8787 | 0.8804 | 26 | 0.9955 | 0.9995 | 0.9960 | 13 | -13 | 0.2842 |
| Henan | 0.9249 | 0.9999 | 0.9250 | 8 | 1.0000 | 1.0000 | 1.0000 | 1 | -7 | 0.0812 |
| Hubei | 0.8907 | 0.9890 | 0.9009 | 18 | 0.9895 | 0.9960 | 0.9934 | 18 | 0 | 0.1109 |
| Hunan | 0.9416 | 0.9912 | 0.9500 | 6 | 0.9974 | 0.9980 | 0.9994 | 10 | 4 | 0.0593 |
| Guangdong | 0.8758 | 0.9999 | 0.8758 | 21 | 1.0000 | 1.0000 | 1.0000 | 1 | -20 | 0.1418 |
| Guangxi | 0.8949 | 0.9940 | 0.9000 | 17 | 0.9822 | 0.9955 | 0.9866 | 21 | 4 | 0.0976 |
| Hainan | 0.8215 | 0.8748 | 0.9435 | 25 | 0.9121 | 0.9369 | 0.9748 | 30 | 5 | 0.1103 |
| Chongqing | 0.8530 | 1.0000 | 0.8530 | 23 | 1.0000 | 1.0000 | 1.0000 | 1 | -22 | 0.1723 |
| Sichuan | 0.7478 | 0.7869 | 0.9477 | 27 | 0.9321 | 0.9474 | 0.9842 | 27 | 0 | 0.2465 |
| Guizhou | 0.9877 | 1.0000 | 0.8977 | 2 | 1.0000 | 1.0000 | 1.0000 | 1 | -1 | 0.0125 |
| Yunnan | 0.8817 | 0.8909 | 0.9894 | 19 | 0.9125 | 0.9397 | 0.9717 | 29 | 10 | 0.0349 |
| Xizang | 1.0000 | 1.0000 | 1.0000 | 1 | 1.0000 | 1.0000 | 1.0000 | 1 | 0 | 0.0000 |
| Shaanxi | 0.7341 | 0.7838 | 0.9355 | 28 | 0.9254 | 0.9350 | 0.9897 | 28 | 0 | 0.2606 |
| Gansu | 0.9592 | 0.9860 | 0.9729 | 3 | 0.9826 | 0.9955 | 0.9870 | 20 | 17 | 0.0244 |
| Qinghai | 0.9105 | 0.9737 | 0.9346 | 14 | 0.9880 | 0.9927 | 0.9953 | 19 | 5 | 0.0851 |
| Ningxia | 0.9040 | 0.9972 | 0.9065 | 16 | 0.9947 | 0.9993 | 0.9953 | 14 | -2 | 0.1003 |
| Xinjiang | 0.9199 | 0.9496 | 0.9685 | 12 | 0.9430 | 0.9958 | 0.9470 | 25 | 13 | 0.0251 |
| Mean | 0.8711 | 0.9573 | 0.9069 | | 0.9749 | 0.9860 | 0.9885 | | | |

all weak DEA effective. In addition, the changes in comprehensive technical efficiency and pure technical efficiency are basically the same, and the insufficiency of comprehensive technical efficiency is mainly caused by insufficient scale efficiency.

**4.2.2 The second stage.** The redundant values of the six input indicators in the three dimensions of digital infrastructure, capital and data platform investment are explained variables. They are explained by six environmental variables in five dimensions: the economic environment, local finance, the industrial structure, technological innovation, and education level. Frontier4.1 is used to perform panel SFA regression (Table 3). In Table 3, the $\gamma$ values of the six regression equations are all greater than 0.8, and the value of the mobile communication network penetration rate, the length of optical cable per square kilometer, the cumulative number of AISs, and the number of agricultural public opinion monitoring platforms are at or

**Table 3. SFA regression results.**

| variable | Slack variable | | | | | |
|---|---|---|---|---|---|---|
| | $I_1$ | $I_2$ | $I_3$ | $I_4$ | $I_5$ | $I_6$ |
| constant | -1.471*(-1.907) | 105.292*** (3.115) | 57.32***{4.561) | 248.451*(1.769) | -5.219*** (-1.910) | -12.319 (-0.729) |
| $E_1$ | -0.057 (1.039) | 0.235*** (6.344) | 0.002*** (3.165) | -0.005*** (4.313) | 0.053*** (3.045) | -0.006*** (-4.726) |
| $E_2$ | 0.248*** (22.246) | 0.022*** (-2.998) | 3.035** (2.011) | 35.022 (-0.724) | 7.899* (1.862) | 0.011** (6.199) |
| $E_3$ | 0.046*** (4.911) | -0.316** (-1.926) | -5.031** (-2.574) | -0.116*** (-5.295) | -0.015*** (-3.446) | -0.013 (-0.589) |
| $E_4$ | -15.111*** (-10.423) | 9.424 (0.593) | -4.826 (-1.136) | -0.187 (-0.407) | -11.026*** (-3.341) | -6.036* (-1.701) |
| $E_5$ | 0.158* (1.678) | 0.002* (1.660) | 0.047* (1.937) | 0.028*** (6.592) | 0.234*** (4.018) | 0.002*** (4.001) |
| $E_6$ | -161.486* (-1.769) | -1887.87*** (-3.894) | -1512.24 (-1.038) | -5123.815*** (-2.599) | -17.818 (1.484) | -1.845*** (-3.007) |
| $\sigma^2$ | 1351.7*** (2.738) | 20292.3*** (13.75) | 199690*** (1653.29) | 216946.8*** (1000.55) | 534.38*** (3.682) | 8497.29*** (4.188) |
| $\gamma$ | 0.979*** (110.99) | 0.94*** (108.27) | 0.898*** (92.782) | 0.834*** (50.073) | 0.989*** (309.86) | 0.956*** (80.567) |
| Maximum likelihood estimator | -629.273 | -995.806 | -1245.480 | -1307.876 | -491.593 | -860.340 |
| LR test | 192.917*** | 233.076*** | 220.777*** | 122.107*** | 265.251*** | 193.074*** |

above 0.9. The likelihood ratio (LR) unilateral test values of the environmental variables were all significant at the 1% level; the coefficients of the influence of various environmental variables on the input slack variables mostly passed the *t*-test. Therefore, the environmental factors and management inefficiencies in the first-stage model disturb the results, and the original input value needs to be adjusted.

Regarding educational level, the coefficient of the compulsory education teacher-student ratio is significantly negative. This result means that a good educational level will effectively reduce the redundancy of investment in the construction of digital villages, and improving the cultural level will directly drive the overall optimization of farmers' digital literacy and have a positive impact on urban-rural balanced development.

Regarding scientific and technological innovation, the number of granted invention patents has a significantly positive effect on the input slack variable of digital village to empower the balanced development of urban and rural areas. This result means that the higher the level of technological innovation is, the greater the redundancy of investment in enabling the balanced development of urban and rural digital villages because the implementation and

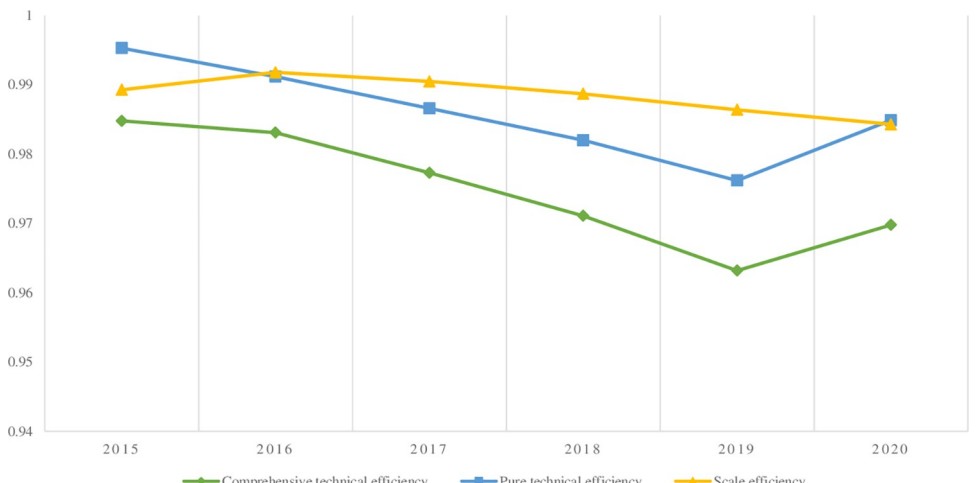

**Fig 2. Efficiency values from 2015 to 2020.**

popularization of high technology in rural areas often takes a longer amount of time and more manpower to publicize, and the short-term impact is not significant.

Regarding the economic environment, per capita GDP has a significant negative impact on $I_1$, $I_4$, $I_6$. Variable input redundancy is weakened, which is conducive to the overall development of urban and rural areas. However, per capita GDP has a significant positive impact on $I_2$, $I_3$, $I_5$. This result is mainly due to the high dependency ratio of the rural population and rural labor. The lack of high-tech talent and the rising overall economic level have caused more investment redundancy in the construction of digital villages. The coefficient of the effect of urbanization level on the input slack variable is significantly positive, indicating that urban agglomeration and expansion and the allocation of rural resources are not optimized at the same time. The probability of rural laborers moving to cities and towns is significantly increasing, resulting in increased input redundancy, which is not conducive to digital villages.

Regarding local finance, fiscal revenue has a positive effect only on the mobile communication network penetration rate slack variable, and for the remaining input slack variables, it has a negative effect, indicating that the increase in local government fiscal revenue can effectively reduce the investment redundancy in the construction of digital villages. At present, China is in the initial stage of constructing digital villages. With government-led promotion as the core, the construction of numerous digital technology facilities in rural areas requires support from a strong fiscal revenue.

Regarding the industrial structure, the proportion of the tertiary industry has a positive impact on the input slack variable fiber optic cable length per square kilometer per capita, and it has a significant negative impact on the remaining input slack variables. With the upgrading of the industrial structure, it is possible for urban and rural areas to realize industrial integration and development with the help of the digital technology.

**4.2.3 The third stage.** The adjusted input variables and original output variables according to Formula (4) repeat the first-stage operation to obtain the efficiency (Table 2). The comprehensive technical efficiency calculated in the third stage to empower the urban-rural balanced development is significantly improved compared with first-stage result. Shandong, Fujian and Shaanxi have the largest increase in comprehensive technical efficiency, all of which are above 25%. The main reason is that the efficiency of these five provinces to empower the urban-rural balanced development is largely affected by environmental and random factors. Gansu, Tianjin, and Xinjiang have the highest overall technical efficiency rankings, rising by 17, 15, and 13 positions, respectively. Chongqing, Guangdong, Shanghai and Shandong's overall technical efficiency rankings decrease the most, dropping by 22, 20, 13, and 13 places, respectively. The comprehensive technical efficiency level of all provinces increases significantly. With the exception of Fujian, the comprehensive technical efficiency of the remaining provinces is above 0.9, and the overall efficiency of each province shows little difference.

Notably, Jiangsu and Zhejiang are developed provinces in China; their first-stage comprehensive technical efficiency figures o are 0.6812 and 0.6209, respectively. After excluding environmental factors and random noise, the comprehensive technical efficiency of the third stage can increase significantly, but Jiangsu and Zhejiang are ranked relative to other provinces that are still in a depression, hence ranking only 26th and 23rd, respectively. There may be two main reasons: (1) during the Thirteenth Five-Year Plan period, developed provinces led by Jiangsu and Zhejiang competed for talent, which in turn, facilitated the settlement of rural senior intellectuals and highly skilled talent in cities, and it has become more difficult to promote rural digital technologies. (2) Although the absolute amount of financial support for agriculture in Jiangsu and Zhejiang is far ahead at the overall regional level, the proportion of financial support for agriculture in fiscal revenue is relatively low, the level of financial support

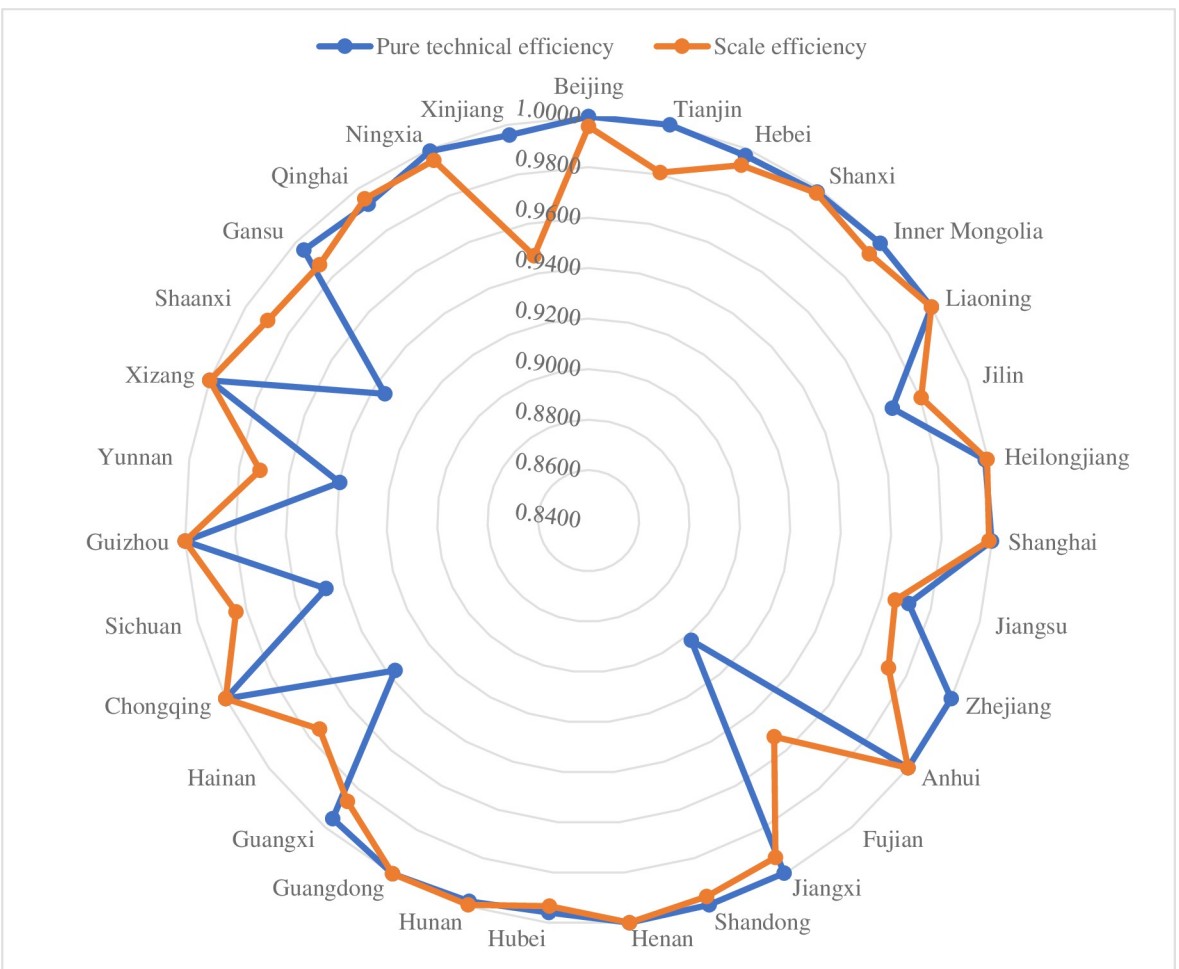

**Fig 3. Radar chart of pure technical efficiency and scale efficiency.**

for scientific and technological innovation is relatively high, and resources and funds are invisible to cities and towns. The efficiency in these two regions is not sufficient.

In Fig 2, the comprehensive technical efficiency of China's digital villages in empowering the balanced development of urban and rural areas from 2015 to 2020 is above 0.96, but the overall trend is declining. Since 2016, pure technical efficiency has been lower than scale efficiency, and the gap between the two has gradually increased. Pure technical efficiency has become a key factor hindering the improvement in comprehensive technical efficiency, changing the pattern of scale efficiency dominance in the first stage and indicating that environmental factors can significantly improve pure technical efficiency.

**4.2.4 Interprovincial heterogeneity.** To further compare the differences in the levels of pure technical efficiency and scale efficiency of each province, we draw a radar chart of each province in the third stage (Fig 3).

In Fig 3, Xinjiang, Shaanxi, Yunnan, Sichuan, Hainan, Fujian, Zhejiang, and Tianjin have large gaps in pure technical efficiency and scale efficiency, indicating that the scale efficiency and pure technical efficiency of the 8 provinces are unbalanced. Among the 31 provinces, only Xinjiang, Zhejiang, and Tianjin have pure technical efficiency higher than scale efficiency, indicating that the scale efficiency of these three regions is the main factor restricting the

**Table 4. Regional distribution of super efficiency.**

| Province | Super efficiency | Type of efficiency | Province | Super efficiency | Type of efficiency |
|---|---|---|---|---|---|
| | | Low efficiency [0.7694,0.9550] | Guangxi | 1.0010 | Medium-high efficiency (1.0005,1.0233] |
| | | | Hubei | 1.0039 | |
| Fujian | 0.8610 | | Hebei | 1.0086 | |
| Hainan | 0.9121 | | Inner Mongolia | 1.0107 | |
| Yunnan | 0.9125 | | Xizang | 1.0149 | |
| Shaanxi | 0.9254 | | Shanxi | 1.0163 | |
| Sichuan | 0.9321 | | Beijing | 1.0174 | |
| Jiangsu | 0.9376 | | Chongqing | 1.0186 | |
| Xinjiang | 0.9486 | | Guangdong | 1.0204 | |
| Jilin | 0.9494 | | Guizhou | 1.0218 | |
| | | | Anhui | 1.0222 | |
| | | | Tianjin | 1.0225 | |
| | | Medium efficiency (0.9550,1.0005] | Shanghai | 1.0249 | High efficiency (1.0233,1.1593] |
| Zhejiang | 0.9815 | | Liaoning | 1.0253 | |
| Gansu | 0.9916 | | Henan | 1.0268 | |
| Shandong | 0.9981 | | Hunan | 1.0300 | |
| Qinghai | 0.9985 | | Heilongjiang | 1.0330 | |
| Jiangxi | 0.9995 | | Ningxia | 1.0437 | |

comprehensive technical efficiency of digital villages in enabling the balanced development of urban and rural areas. In provinces such as Shaanxi, Yunnan, Sichuan, Hainan, Fujian and Jilin, scale efficiency is significantly higher than pure technical efficiency, and improving pure technical efficiency has become the main way to improve overall technical efficiency.

The efficiency measured by the basic DEA model will result in the efficiency value of multiple DMUs being 1, which is not conducive to presenting the rankings of provinces. Accordingly, the adjusted input variable and initial output variable are measured as a super-efficiency value, and all efficiency values are sorted by quartile, forming four grades: high efficiency, medium-high efficiency, medium efficiency and low efficiency (Table 4).

Table 4 shows that the super-efficiency value of 18 provinces is greater than 1, meaning that these provinces have middle-high and high efficiency segments. Moreover, the DEA of these 18 provinces remains effective after increasing the input of a certain scale on the current input–output scale, but the DEA of the provinces with low and medium efficiency is still weak and ineffective.

**4.2.5 Malmquist exponential decomposition.** To observe the dynamic changes in digital villages to empower urban-rural equilibrium development efficiency in different provinces, this paper uses the global reference Malmquist index (total factor productivity) to measure the changes in the technological level of different provinces. The Malmquist index indicates the change in efficiency caused by the common trend of internal and external factors. The technological progress index is the change in the efficiency of urban-rural equilibrium development driven by external factors such as technological innovation and policy transmission. The Malmquist index is equal to comprehensive technical efficiency multiplied by the technological progress index. A Malmquist index greater than 1 indicates that total factor productivity is increasing, which is called efficiency growth. If the Malmquist index is less than 1, then it indicates that total factor productivity is showing a downward trend, which is called inefficiency Using MAXDEA software, according to the third-stage model calculation, the changes in and

**Table 5. Malmquist index of each province and its decomposition.**

| | Malmquist index | Pure technical efficiency change | Scale efficiency changes | Technological progress index | Comprehensive technical efficiency | Rank |
|---|---|---|---|---|---|---|
| Beijing | 1.0041 | 1.2353 | 0.9688 | 0.9442 | 1.0691 | 6 |
| Tianjin | 0.9582 | 1.0304 | 0.9451 | 0.9846 | 0.9737 | 31 |
| Hebei | 0.9931 | 0.9883 | 1.0048 | 1.0005 | 0.9934 | 21 |
| Shanxi | 0.9879 | 1.0043 | 0.9965 | 0.9885 | 1.0005 | 25 |
| Inner Mongolia | 1.0016 | 1.0161 | 0.9987 | 0.9875 | 1.0148 | 8 |
| Liaoning | 0.9936 | 1.0078 | 1.0025 | 0.9838 | 1.0102 | 19 |
| Jilin | 0.981 | 0.9746 | 1.0311 | 0.9827 | 1.002 | 27 |
| Heilongjiang | 0.9979 | 1.1031 | 0.9215 | 0.9865 | 1.0114 | 14 |
| Shanghai | 0.9849 | 1.0055 | 1.0156 | 0.9668 | 1.0194 | 26 |
| Jiangsu | 0.9929 | 1.02 | 0.992 | 0.9886 | 1.0044 | 22 |
| Zhejiang | 0.9795 | 1.0212 | 0.9714 | 0.9912 | 0.9883 | 28 |
| Anhui | 1.0083 | 1.2137 | 0.8846 | 0.9746 | 1.0352 | 3 |
| Fujian | 0.9609 | 0.9698 | 1.04 | 0.9566 | 1.0055 | 30 |
| Jiangxi | 0.9942 | 1.0007 | 0.9892 | 1.0068 | 0.9881 | 17 |
| Shandong | 0.9975 | 1.0097 | 0.9983 | 0.9933 | 1.0083 | 15 |
| Henan | 1.0007 | 1.0551 | 0.9816 | 0.9667 | 1.0351 | 11 |
| Hubei | 0.9932 | 1.0283 | 0.9684 | 1.0061 | 0.9876 | 20 |
| Hunan | 1.0159 | 1.0432 | 0.9789 | 1.0036 | 1.0199 | 2 |
| Guangdong | 0.9965 | 0.9817 | 1.0362 | 0.9864 | 1.0106 | 16 |
| Guangxi | 0.9763 | 1.0698 | 0.9285 | 0.9975 | 0.9808 | 29 |
| Hainan | 1.0018 | 1.0306 | 1.0079 | 0.9651 | 1.0383 | 7 |
| Chongqing | 0.9994 | 0.9904 | 0.9934 | 1.0169 | 0.9834 | 13 |
| Sichuan | 0.9926 | 0.9912 | 0.9985 | 1.0035 | 0.9899 | 24 |
| Guizhou | 1.0012 | 1.0012 | 1.0001 | 1.0033 | 1.0002 | 10 |
| Yunnan | 0.9939 | 0.9909 | 1.0073 | 0.9961 | 0.998 | 18 |
| Xizang | 1 | 1.0288 | 1.0165 | 0.9579 | 1.0452 | 12 |
| Shaanxi | 1.0013 | 1.0061 | 0.9998 | 0.9959 | 1.0058 | 9 |
| Gansu | 1.0077 | 1.1001 | 1.0246 | 0.9723 | 1.0374 | 4 |
| Qinghai | 0.9929 | 0.9953 | 0.9999 | 0.9987 | 0.9946 | 23 |
| Ningxia | 1.0071 | 0.8626 | 1.1894 | 1.0108 | 0.9956 | 5 |
| Xinjiang | 1.0229 | 1.0142 | 1.0099 | 0.9992 | 1.0238 | 1 |

decomposition of the Malmquist index of the average digital village in 31 Chinese provinces from 2015 to 2020 are shown in Table 5.

Table 5 shows that the total factor productivity values of all provinces in the country are relatively high. Among them, 11 provinces, such as Xinjiang, Hunan, and Anhui, are characterized by efficiency growth, which shows that there is a certain degree of growth in total factor productivity in these 11 provinces. The Malmquist index of the remaining 20 provinces is between 0.95 and 1; that is, they are inefficient. Based on the decomposition of the Malmquist index, Tianjin, Guangxi, Chongqing, Hubei and Jiangxi are the five provinces with the worst overall technical efficiency changes, and Beijing, Xizang, Hainan, Gansu and Anhui are the provinces with the best overall technical efficiency changes. There are 8 provinces where the value of the technological progress index is greater than 1. Among them, Chongqing, Ningxia and Jiangxi have the most significant technological progress, and the technological progress index generally has a low value, becoming the most important factor restricting the efficiency

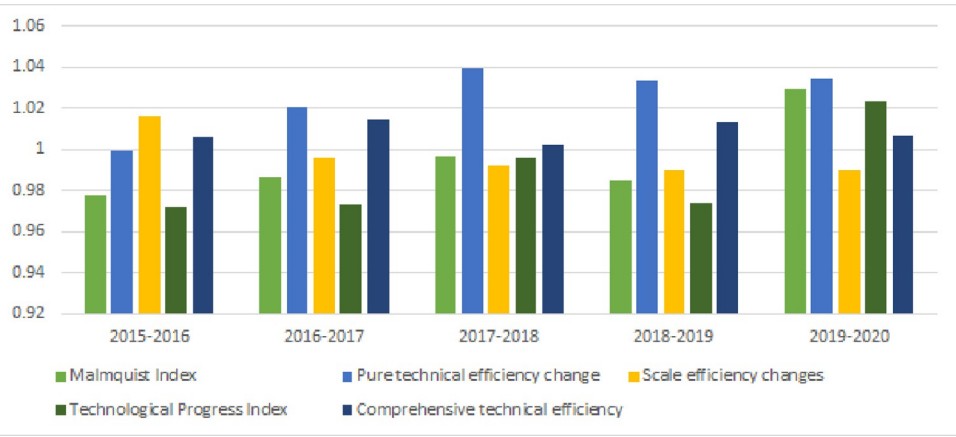

**Fig 4. Malmquist index and decomposition, 2015–2020.**

of digital villages in empowering the balanced development of urban and rural areas. Only 10 provinces have a technological progress index higher than the pure technological efficiency change: Ningxia, Qinghai, Yunnan, Guizhou, Sichuan, Chongqing, Guangdong, Jiangxi, Jilin and Hebei.

The changes in the China Malmquist index from 2015 to 2020 and its decomposition are shown in Fig 4. The figure shows that the values of the Malmquist index from 2015 to 2019 are all less than 1, indicating that the efficiency of China's digital villages in empowering urban-rural balanced development shows a law of diminishing marginal utility. The Malmquist index was greater than 1 in 2019–2020 period, which was affected by the COVID-19 pandemic. The whole country has better realized the overall development of urban and rural areas. During the 2015–2018 period, total factor productivity showed an increasing trend, but it showed a down-ward trend in the 2018–2019 period. The main reason is that the central government issued the "Digital Village Development Strategy Outline" in May 2019, but provinces could not clearly grasp the construction goals and paths beforehand, which caused insufficient invest-ment. Decomposition data based on the Malmquist index show that, in relative terms, the value of scale efficiency change is the lowest, and its growth rate is negative in the 2015–2019 period. The technological progress index has generally shown a growth trend, and the growth rate has been increasing year by year. The growth of the technological progress index has driven the increase in total factor productivity, and it has well hedged the result of the decline in total factor productivity driven by the decline in scale efficiency. The technological progress index affects the key factor of total factor productivity, and technological progress has become the best path for digital villages to empower the balanced development of urban and rural areas to improve quality and efficiency.

## 5 Conclusion

### 5.1 Discussion

First, digital education empowers rural entities with digital literacy, digital platforms empower rural entities with digital technology utilization, and digital infrastructure empowers rural entities with digital behaviors. All of this can help digital villages fully empower rural entities and bridge the urban-rural income gap, and bring to an end the differences in urban-rural consumption and employment. Digital education, digital platforms, and digital infrastructure promote the integration of urban-rural economies by empowering the precision of the

agricultural production system, the scale of agricultural operations, and the efficiency of the management system. They also effectively fill the gap in equal urban-rural development and promote the equalization of urban-rural education, medical care, and governance by empowering the digitalization of rural planning, public services, and governance.

Second, under the input–output theoretical framework, six indicators based on the three perspectives of digital infrastructure investment, capital investment, and data platform investment are selected as input variables. Equilibrium in the urban-rural economy, employment, consumption, social security and digital living environment are output variables. The evaluation index system for the efficiency of digital villages in empowering urban-rural balanced development is constructed, and five environmental variables, i.e., the economic environment, local finance, the industrial structure, technological innovation and educational level, are selected.

Third, based on the empirical analysis of the three-stage DEA-Malmquist model, we conclude the following: (1) Comprehensive technical efficiency, pure technical efficiency, and scale efficiency are all weak DEA effect, and insufficient scale efficiency is the main factor. (2) The SFA regression results show that environmental factors and management inefficiencies have disturbed the efficiency of the first stage, making it necessary to perform SFA regression. Educational leve;, local finance, and industrial structure optimization can effectively promote urban-rural balanced development; in contrast, technological innovation and the urbanization level hinder balanced development. Due to the relatively high dependency of the rural population, the lack of high-tech talent, and the difficulty of implementing new technologies in rural areas, the influence of the economic environment on efficiency is not completely uniform. (3) After removing environmental factors and random noise, the comprehensive technical efficiency of 31 prefecture-level cities showed a rising trend, and the factors affecting the comprehensive technical efficiency were different. However, based on the temporal dimension, comprehensive technical efficiency decreases year by year. The super efficiency values are obviously different, and 5 regions are at the frontier of technical efficiency. (4) Based on the Malmquist index and its decomposition, only 11 provinces in China exhibit total factor productivity growth. Meanwhile, the efficiency of digital villages in enabling balanced urban-rural development shows diminishing marginal utility.

To better realize the efficiency of digital villages in empowering balanced urban-rural development, we propose many improvement measures. (1) The government should strengthen investment in and the operation of digital platform construction. With the help of mini-programs and official accounts, AISs have accurately deployed resources for agriculture and farmers in rural areas. However, at the level of national construction, the number of visits and the participation rate of rural subjects still need to be improved, and the effect of helping farmers in employment and income generation has not been fully demonstrated. Therefore, it is necessary to rely on the advantages of data to achieve the vertical extension of the agricultural industry chain and the deep integration of agriculture and culture. In the construction of digital platforms, window service functions are optimized to avoid homogeneity and improve accuracy and convenience. The operation of the digital platform can effectively avoid investment redundancy, guide rural entities, industries and governments to use the platform, and improve the degree of information connection. (2) The government should effectively extend the whole "Internet + Agriculture" industry chain. In the context of the global pandemic, the digital economy is the current new economic growth point. It is necessary and feasible to make full use of internet platforms to vigorously promote green organic agricultural products and to use digital technology to accurately distribute information. At the front end, the supply and demand matching link between agricultural products and final consumers is realized; at the back end, the logistics and after-sales service systems for agricultural products are

continuously optimized. This can increase the operating income of farmers and further promote the upgrading of rural consumption.

## 5.2 Contributions and limitations of this paper

First, this paper builds a theoretical transmission mechanism for digital villages to empower the balanced development of urban and rural areas, based on the input-output framework. We construct an evaluation index system for the efficiency of digital villages in empowering urban-rural balanced development. Empirical tests confirm the differences between different provinces in China.

Because the construction of digital villages is in the initial stage of development, many of the investment indicators for measuring the construction of digital villages are not quantifiable, and it is difficult to obtain data, which shows that some aspects of digital villages empowering the balanced development of urban and rural areas have not yet been comprehensively realized for scientific assessment. With the continuous implementation of the digital village construction plan by governments at all levels, the efficiency measurement for enabling the balanced development of urban and rural areas will be more precise in the future.

## Acknowledgments

The authors would like to thank all referees and a co-editor for their comments and suggestions.

## Author Contributions

**Conceptualization:** Huawei Niu.

**Data curation:** YiFeng Wang.

**Funding acquisition:** Huawei Niu.

**Investigation:** YiFeng Wang.

**Methodology:** Lingling Cao.

**Resources:** Lingling Cao, Huawei Niu.

**Software:** Lingling Cao.

**Supervision:** YiFeng Wang.

**Validation:** YiFeng Wang.

**Writing – original draft:** Lingling Cao.

**Writing – review & editing:** Huawei Niu.

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
