## [Decision Letter · Decision Letter 0]

11 May 2022

PONE-D-22-02890The Utility Analysis of Digital Village Empowering Balanced Urban-Rural Development Based on the Three-stage DEA-Malmquist ModelPLOS ONE

Dear Dr. Cao,

Thank you for submitting your manuscript to PLOS ONE. After careful consideration, we feel that it has merit but does not fully meet PLOS ONE’s publication criteria as it currently stands. Therefore, we invite you to submit a revised version of the manuscript that addresses the points raised during the review process.

We look forward to receiving your revised manuscript.

Kind regards,

Hao Xue

Academic Editor

PLOS ONE

Journal Requirements:

Reviewers' comments:

Reviewer's Responses to Questions

**Comments to the Author**

1. Is the manuscript technically sound, and do the data support the conclusions?

Reviewer #1: Yes

Reviewer #2: Yes

2. Has the statistical analysis been performed appropriately and rigorously? 

Reviewer #1: Yes

Reviewer #2: Yes

3. Have the authors made all data underlying the findings in their manuscript fully available?

Reviewer #1: Yes

Reviewer #2: Yes

4. Is the manuscript presented in an intelligible fashion and written in standard English?

Reviewer #1: No

Reviewer #2: Yes

5. Review Comments to the Author

Reviewer #1: The Utility Analysis of Digital Village Empowering Balanced Urban-rural Development Based on the three-stage dea-Malmquist Model, it is of great significance.

In general, Experiments, statistics, And other analyses are performed to a high technical standard and are described in sufficient detail; Its conclusions are presented in an appropriate fashion and are supported by the data. It is suggested to be slightly revised and then accepted for publication.

The following are some suggestions for minor modifications:

1. What is the “mobile divide” and “digital divide”? (Page 2) What are their definitions?

There should be an explanation for these related concepts.

2.It is suggested to further add "discussion" and "contribution and limitation of this paper" in the "Conclusion" section.

3.The language needs further polishing. It is recommended to invite a native English speaker to help polish the language.

Reviewer #2: This is a well written article discussing how the digital development has affected rural development and facilitate the urban-rural equilibrium. It is well organised and the method is solid. I have one concern about the literature review. I understand that this is a more macro level discussion. But some case studies should be better reviewed, e.g. the Taobao villages. I notice that there are quite many studies on Taobao villages. Some of them should be included in your literature review. The following references can be useful:

1)Informality and rural industry: Rethinking the impacts of E-Commerce on rural development in China. Journal of Rural Studies

2) E-Commerce and Taobao villages. A promise for China's rural development? China Perspective

Also, the figure 1 is not very clear. In section 2, there is only one reference. I suggest you add more when introduce your framework.

6. PLOS authors have the option to publish the peer review history of their article (what does this mean?). If published, this will include your full peer review and any attached files.

Reviewer #1: No

Reviewer #2: No

---

## [Author Response · Author response to Decision Letter 0]

27 May 2022

PONE-D-22-02890

Utility analysis of digital village to empower balanced urban-rural development based on the three-stage DEA-Malmquist model

Journal Requirements:

Answer: Thank you for your advice. We have completely revised the format of the article. We are sure that the manuscript meets PLOS ONE's style requirements. The file name also meets the requirements.

Answer: Thank you for your comment. The funding support information has been fully revised. We are sure that we provide the correct grant numbers for the awards. The details as follows:

Funding: This research was supported by the National Natural Science Foundation of China (71871120)，the Excellent Social Science Application Engineering Projects of Jiangsu Province (21SYB-091) and the “Blue Project” of Jiangsu University. The funders had no role in the study design, data collection and analysis, decision to publish, or preparation of the manuscript.

3.Please review your reference list to ensure that it is complete and correct. If you have cited papers that have been retracted, please include the rationale for doing so in the manuscript text, or remove these references and replace them with relevant current references. Any changes to the reference list should be mentioned in the rebuttal letter that accompanies your revised manuscript. If you need to cite a retracted article, indicate the article’s retracted status in the References list and also include a citation and full reference for the retraction notice.

Answer: Thank you for your comments. All references have been revised based on the journal’s requirements. We ensured that all references were correct and complete. Based on the reviewers' comments, we added 8 references, replaced one reference, and deleted one reference. The original reference “Digital Technologies in Agriculture and Rural Areas Status Report” was replaced by the new reference “A review of social science on digital agriculture, smart farming and agriculture 4.0: New contributions and a future research agenda”.

References: 

1.Lee JH, Kim J. Socio-demographic gaps in mobile use, causes, and consequences: a multi-group analysis of the mobile divide model. Information Communication & Society. 2014;17(8):917-936.

https://doi.org/10.1080/1369118X.2013.860182

2.Rice RE, Katz JE. Comparing internet and mobile phone usage: Digital divides of usage, adoption, and dropouts. Telecommunications Policy.2003;27(8):597–623. https://doi.org/10.1016/S0308-5961(03)00068-5

3.Pylianidis C, Osinga S, Athanasiadis IN. Introducing digital twins to agriculture. Computers and Electronics in Agriculture.2021;184(4):105942.

https://doi.org/10.1016/j.compag.2020.105942

4.Yang RJ, Cao YP. On the tension between rural digital empowerment and digital divide and its resolution. Journal of Nanjing Agricultural University (Social Science Edition). 2021;21(5),31-40.https://doi.org/10.19714/j.cnki.1671-7465.2021.0070.

5.Basso B, Antle J. Digital agriculture to design sustainable agricultural systems. Nature Sustainability.2020;3(4):254–256. https://doi.org/10.1038/s41893-020-0510-0

6.Rijswijk K, Klerkx L, Turner JA, Digitalisation in the New Zealand agricultural knowledge and innovation system：Initial understandings and emerging organisational responses to digital agriculture. NJAS-Wageningen Journal of Life Sciences. 2019;90-91(5):100313. https://doi.org/10.1016/j.njas.2019.100313

7.Wang M. Possible adoption of precision agriculture for developing countries at the threshold of the new millennium. Computers and Electronics in Agriculture. 2001;30(1-3):45-50. https://doi.org/10.1016/S0168-1699(00)00154-X

8.Hennessy T，Lapple D, Moran B. The digital divide in farming: A problem of access or engagement? Applied Economic Perspectives and Policy.2016; 38(3):474-491. https://doi.org/10.1093/aepp/ppw015

9.Runck BC, Joglekar A, Silverstein KAT, Chan-Kang C, Pardey PG, Wilgenbusch, JC. Digital agriculture platforms: Driving data-enabled agricultural innovation in a world fraught with privacy and security concerns. Agronomy journal. 2021. https://doi.org/10.1002/agj2.20873

10.Deichmann U，Goyal A，Mishra D. Will digital technologies transform agriculture in developing countries? Agricultural Economics. 2016;47:21-33. 

https://doi.org/10.1111/agec.12300

11.Rijswijk K, Klerkx L, Bacco M, Bartolini F, Bulten E, Debruyne L, et al. Digital transformation of agriculture and rural areas: A socio-cyber-physical system framework to support responsibilisation. Journal of Rural Studies. 2021;85:79-90. https://doi.org/10.1016/j.jrurstud.2021.05.003

12.Klerkx L, Jakku E, Labarthe P. A review of social science on digital agriculture, smart farming and agriculture 4.0: New contributions and a future research agenda. NJAS-Wageningen Journal of Life Sciences.2019;90-91:100315. https://doi.org/10.1016/j.njas.2019.100315

13.Li AHF. E-commerce and Taobao Villages: A Promise for China's Rural Development?. China Perspectives. 2017; (3):57-62. 

https://doi.org/10.4000/chinaperspectives.7423

14.Tang W , Zhu J . Informality and rural industry: Rethinking the impacts of E-Commerce on rural development in China. Journal of Rural Studies.2020;75:20-29. https://doi.org/10.1016/j.jrurstud.2020.02.010

15.Qi JQ, Zheng XY, Guo HD. The formation of Taobao villages in China. China Economic Review. 2019;53:106-127. 

https://doi.org/10.1016/j.chieco.2018.08.010

16.Wang WY, Li JS, Liu WM, Liu ZK. Integrated computational materials engineering for advanced materials: A brief review. Computational Materials Science.2019;158:42-48. https://doi.org/10.1016/j.commatsci.2018.11.001

17.Salemink K, Strijker D, Bosworth G. The Community Reclaims Control? Learning Experiences from Rural Broadband Initiatives in the Netherlands. Sociologia Ruralis.2017;57:555-575. https://doi.org/10.1111/soru.12150

18.Sidibe A, Olabisi LS, Doumbia H, Toure K, Niamba CA. Barriers and enablers of the use of digital technologies for sustainable agricultural development and food security. Elementa-Science of the Anthropocene.2021;9(1). https://doi.org/10.1525/elementa.2020.00106

19.Xie L , Luo BL, Zhong WJ. How Are Smallholder Farmers Involved in Digital Agriculture in Developing Countries: A Case Study from China. Land.2021;10(3):245. https://doi.org/10.3390/land10030245

20.Prause L, Digital Agriculture and Labor: A Few Challenges for Social Sustainability. Sustainability.2021;13(11):5980. 

https://doi.org/10.3390/su13115980

21.Janowski T. Implementing Sustainable Development Goals with Digital Government-Aspiration-capacity gap. Government Information Quarterly.2016; 33(4): 603-613. https://doi.org/10.1016/j.giq.2016.12.001

22.Janssen M, Haiko V. Adaptive governance: Towards a stable，accountable and responsive government. Government Information Quarterly.2016;33(1):1-5. https://doi.org/10.1016/j.giq.2016.02.003

23.Liu SB, Guo LQ, Webb H, Ya X, Chang X. Internet of things monitoring system of modern eco-agriculture based on cloud computing. IEEE Access.2019;7:37050-37058. https://doi.org/10.1109/ACCESS.2019.2903720

24.Loures L, Chamizo A, Ferreira P, Loures A, Castanho R, Panagopoulos T. Assessing the Effectiveness of Precision Agriculture Management Systems in Mediterranean Small Farms. Sustainability. 2020;12(9):3765. 

https://doi.org/10.3390/su12093765

25.Hrustek L. Sustainability driven by agriculture through digital transformation. Sustainability. 2020;12(20):8596. https://doi.org/ 10.3390/su12208596

26.Ramanadhan S, Ganapathy K, Nukala L, Rajagopalan S, Camillus, JC. A model for sustainable, partnership-based telehealth services in rural India: An early process evaluation from Tuver village, Gujarat. PloS one.2022;17(1): e0261907 https://doi.org/10.1371/journal.pone.0261907

27.Aho K, Derryberry D, Peterson T. Model selection for ecologists: the worldviews of AIC and BIC. Ecology, 2014,95(3): 631-636. https://doi.org/10.1890/13-1452.1

 

Review Comments to the Author

Reviewer #1: The Utility Analysis of Digital Village Empowering Balanced Urban-rural Development Based on the three-stage DEA-Malmquist Model, it is of great significance.

In general, Experiments, statistics, and other analyses are performed to a high technical standard and are described in sufficient detail; Its conclusions are presented in an appropriate fashion and are supported by the data. It is suggested to be slightly revised and then accepted for publication.

The following are some suggestions for minor modifications:

1. What is the “mobile divide” and “digital divide”? (Page 2) What are their definitions?

There should be an explanation for these related concepts.

Answer: Thank you for your comments. We have given the definitions of the two proper terms "mobile divide" and "digital divide" in the footnotes of the article. We also added two references. The details as follows: 

“Mobile divide”: Currently, a mobile phone is an important social communication tool and a multifunctional medium. Thus, inequality with regard to possession and use of mobile media creates gaps among different groups, known as the ‘Mobile divide’. 

“Digital divide”: It is defined as the disparity between 'the information-rich' and 'the information-poor’, especially in access to new information technology such as personal computers (PC) and the internet. 

References:

1. Lee JH, Kim J. Socio-demographic gaps in mobile use, causes, and consequences: a multi-group analysis of the mobile divide model. Information Communication & Society. 2014;17(8):917-936.

https://doi.org/10.1080/1369118X.2013.860182

2. Rice RE, Katz JE. Comparing internet and mobile phone usage: Digital divides of usage, adoption, and dropouts. Telecommunications Policy.2003;27(8): 597–623. https://doi.org/10.1016/S0308-5961(03)00068-5

2.It is suggested to further add "discussion" and "contribution and limitation of this paper" in the "Conclusion" section.

Answer: Thank you for your comments. In the Conclusion section, we further enrich the discussion and contributions and limitations of this paper. The details as follows: 

5 Conclusion

5.1 Discussion

First, digital education empowers rural entities with digital literacy, digital platforms empower rural entities with digital technology utilization, and digital infrastructure empowers rural entities with digital behaviors. All of this can help digital villages fully empower rural entities and bridge the urban-rural income gap, and bring to an end the differences in urban-rural consumption and employment. Digital education, digital platforms, and digital infrastructure promote the integration of urban-rural economies by empowering the precision of the agricultural production system, the scale of agricultural operations, and the efficiency of the management system. They also effectively fill the gap in equal urban-rural development and promote the equalization of urban-rural education, medical care, and governance by empowering the digitalization of rural planning, public services, and governance.

Second, under the input–output theoretical framework, six indicators based on the three perspectives of digital infrastructure investment, capital investment, and data platform investment are selected as input variables. Equilibrium in the urban-rural economy, employment, consumption, social security and digital living environment are output variables. The evaluation index system for the efficiency of digital villages in empowering urban-rural balanced development is constructed, and five environmental variables, i.e., the economic environment, local finance, the industrial structure, technological innovation and educational level, are selected.

Third, based on the empirical analysis of the three-stage DEA-Malmquist model, we conclude the following: (1) Comprehensive technical efficiency, pure technical efficiency, and scale efficiency are all weak DEA effect, and insufficient scale efficiency is the main factor. (2) The SFA regression results show that environmental factors and management inefficiencies have disturbed the efficiency of the first stage, making it necessary to perform SFA regression. Educational leve;, local finance, and industrial structure optimization can effectively promote urban-rural balanced development; in contrast, technological innovation and the urbanization level hinder balanced development. Due to the relatively high dependency of the rural population, the lack of high-tech talent, and the difficulty of implementing new technologies in rural areas, the influence of the economic environment on efficiency is not completely uniform. (3) After removing environmental factors and random noise, the comprehensive technical efficiency of 13 prefecture-level cities showed a rising trend, and the factors affecting the comprehensive technical efficiency were different. However, based on the temporal dimension, comprehensive technical efficiency decreases year by year. The super efficiency values are obviously different, and 5 regions are at the frontier of technical efficiency. (4) Based on the Malmquist index and its decomposition, only 11 provinces in China exhibit total factor productivity growth. Meanwhile, the efficiency of digital villages in enabling balanced urban-rural development shows diminishing marginal utility.

To better realize the efficiency of digital villages in empowering balanced urban-rural development, we propose many improvement measures. (1) The government should strengthen investment in and the operation of digital platform construction. With the help of mini-programs and official accounts, AISs have accurately deployed resources for agriculture and farmers in rural areas. However, at the level of national construction, the number of visits and the participation rate of rural subjects still need to be improved, and the effect of helping farmers in employment and income generation has not been fully demonstrated. Therefore, it is necessary to rely on the advantages of data to achieve the vertical extension of the agricultural industry chain and the deep integration of agriculture and culture. In the construction of digital platforms, window service functions are optimized to avoid homogeneity and improve accuracy and convenience. The operation of the digital platform can effectively avoid investment redundancy, guide rural entities, industries and governments to use the platform, and improve the degree of information connection. （2）The government should effectively extend the whole “Internet + Agriculture” industry chain. In the context of the global pandemic, the digital economy is the current new economic growth point. It is necessary and feasible to make full use of internet platforms to vigorously promote green organic agricultural products and to use digital technology to accurately distribute information. At the front end, the supply and demand matching link between agricultural products and final consumers is realized; at the back end, the logistics and after-sales service systems for agricultural products are continuously optimized. This can increase the operating income of farmers and further promote the upgrading of rural consumption.

5.2 Contributions and limitations of this paper 

First, this paper builds a theoretical transmission mechanism for digital villages to empower the balanced development of urban and rural areas, based on the input-output framework. We construct an evaluation index system for the efficiency of digital villages in empowering urban-rural balanced development. Empirical tests confirm the differences between different provinces in China.

Because the construction of digital villages is in the initial stage of development, many of the investment indicators for measuring the construction of digital villages are not quantifiable, and it is difficult to obtain data, which shows that some aspects of digital villages empowering the balanced development of urban and rural areas have not yet been comprehensively realized for scientific assessment. With the continuous implementation of the digital village construction plan by governments at all levels, the efficiency measurement for enabling the balanced development of urban and rural areas will be more precise in the future.

3.The language needs further polishing. It is recommended to invite a native English speaker to help polish the language.

Answer: Thank you for your comment. AJE edited the language of the paper. The following proof is provided:

 

Reviewer #2: This is a well written article discussing how the digital development has affected rural development and facilitate the urban-rural equilibrium. It is well organized and the method is solid. I have one concern about the literature review. I understand that this is a more macro level discussion. But some case studies should be better reviewed, e.g. the Taobao villages. I notice that there are quite many studies on Taobao villages. Some of them should be included in your literature review. The following references can be useful:

1) Informality and rural industry: Rethinking the impacts of E-Commerce on rural development in China. Journal of Rural Studies

2) E-Commerce and Taobao villages. A promise for China's rural development? China Perspective

Answer: Thank you for your advice. The references you provided have made our discussion more thorough and specific. Thank you again. In the literature review section, we have added 3 references to Taobao Villages on the balanced development of urban and rural areas. The specific modifications are as follows:

Third, the integration of digital technology and agricultural economic development can help improve rural economic development and the knowledge and capabilities of farmers, and it can promote agricultural upgrading, rural progress and farmer development in a comprehensive manner [12]. Taobao Village, which have undoubtedly brought economic prosperity to rural areas, are regarded by both academia and the government as an effective means of revitalizing rural areas and narrowing the rural–urban gap [13-15]. The scale and accumulation effect of the “digital economy plus industrial layout and supply chain extension” are conducive to increasing the economic added value of agricultural products and bridging the potential economic gap between urban and rural imbalances [6,16]. Regarding the implementation of urban-rural policies, digital technology drives the dilution of regional characteristics. In addition, the borderless information collection ability promotes the integration of urban and rural areas [17]. A new dual-center policy system of “rural characteristic elements plus urban advantage elements” has been formed [18]. This system will help prevent the “migration boom” and promote the return of resources, technology, and talent to agricultural and rural areas. Digital technology injects new momentum into the development of cultural interconnection between urban and rural areas, helps to fully realize the barrier-free sharing of urban and rural culture, and realizes the parallel complementation and integration of multiple urban and rural cultures [19]. For rural governance, digital technology is used as a driving force to empower the field of rural governance, stimulate villagers’ initiative and build a team of professional governance talent [20]. Doing so requires the integration of shared data resources to realize an “online plus offline, technology and system” new digital rural public service system across regions and departments [21]. For example, the emergence of e-commerce associations in Taobao Villages suggests a larger but contained space for rural e-tailers’ participation in public affairs, leading to a new party-state corporatist mode of rural governance [14].

References:

13. Li AHF. E-commerce and Taobao Villages: A Promise for China's Rural Development?. China Perspectives. 2017; (3):57-62.

https://doi.org/10.4000/chinaperspectives.7423

14. Tang W , Zhu J . Informality and rural industry: Rethinking the impacts of E-Commerce on rural development in China. Journal of Rural Studies.2020;75:20-29. https://doi.org/10.1016/j.jrurstud.2020.02.010

15. Qi JQ, Zheng XY, Guo HD. The formation of Taobao villages in China. China Economic Review. 2019;53:106-127. https://doi.org/10.1016/j.chieco.2018.08.010

Also, the figure 1 is not very clear. 

Answer: Thank you for your advice. We have modified Figure 1 and provided its original image in the attachment. Above Figure 1, we have added generalizations.

Based on the input-output framework, we comprehensively identified various transmission channels for digital villages to enable balanced urban and rural development (Fig 1).

In section 2, there is only one reference. I suggest you add more when introduce your framework.

Answer: Thank you for your advice. In section 2, we added 8 citations and 3 new references. The details are as follows:

References:

23. Liu SB, Guo LQ, Webb H, Ya X, Chang X. Internet of things monitoring system of modern eco-agriculture based on cloud computing. IEEE Access.2019;7:37050-37058. https://doi.org/10.1109/ACCESS.2019.2903720

24. Loures L, Chamizo A, Ferreira P, Loures A, Castanho R, Panagopoulos T. Assessing the Effectiveness of Precision Agriculture Management Systems in Mediterranean Small Farms. Sustainability. 2020;12(9):3765. 

https://doi.org/10.3390/su12093765

26. Ramanadhan S, Ganapathy K, Nukala L, Rajagopalan S, Camillus, JC. A model for sustainable, partnership-based telehealth services in rural India: An early process evaluation from Tuver village, Gujarat. PloS one.2022;17(1): e0261907 https://doi.org/10.1371/journal.pone.0261907

2.1 Empowering agricultural subjects

2.1.1 Digital education empowers digital literacy

Human capital theory confirms that digital technology training for rural subjects can effectively stimulate farmers' ability and willingness to use digital technology and strengthen the technical thinking and modernization consciousness of rural subjects [22] Comprehensive coverage of the whole process and comprehensive digital education should be carried out for rural subjects to promote the enthusiasm and initiative of those who rely on agriculture, and those who return to their hometowns to start businesses and new agricultural operation projects. The popularization of digital education comprehensively inspires the digital universe, and the social, creative and safety literacy of rural subjects. The digital application of rural subjects continues to extend vertically and horizontally, thus driving the continuous growth in and quality of rural internet consumption and narrowing the gap in urban-rural consumption. Multiple channels should be built to fully cover the learning needs of rural subjects and to promote the breadth and depth of rural subjects' direct participation in digital life.

2.1.2 Digital platforms enable the utilization of digital technologies

At the production end of farming and returning entrepreneurial groups, farmers use new social media platforms with a low threshold to promote agricultural and sideline products through multiple channels and accurately adjust sales strategies based on market feedback so that they can improve their ability to connect with the market, expand sales and increase income. Improving the human capital of typical rural subjects has the spillover effect and knowledge diffusion effect of driving the demonstration effect [4]. One person drives a group of people, and one region drives the surrounding region or even infinitely replicates beyond the regional boundary, which has a great driving effect on rural employment.

For new agricultural operators, building a monitoring cloud platform system can help accurately predict market demand and improve quality and efficiency [23]. A digital platform for the circulation of agricultural and sideline products organically connects farmers, middlemen, dealers and other entities to fully share data and information, which is conducive to reducing operating costs. Agglomeration using platforms helps realize information, and resources. Moreover, the information asymmetry of barriers will be broken; thus, using its scale advantage, the platform will constantly attract market participation, the realization of agricultural products and the efficient matching of market supply and demand and increase added value of agriculture.

2.1.3 Digital infrastructure empowers digital behavior

The construction of rural digital infrastructure is conducive to realizing the accessibility and equality of rural subjects' information, blocking the intergenerational transmission of the lack of information ability among rural subjects, and comprehensively stimulating potential digital demand and data acquisition ability [8]. Digital education can empower rural subjects with digital literacy and guide rural subjects in an orderly manner to consume internet finance, online travel and other fields of deep internet application.

2.2 Empowering the agricultural industry

2.2.1 Empowering the precision of the agricultural production system

Digital technology empowers agricultural production systems to achieve the integration of agriculture and other industries. First, it promotes the application of the agriculture and forestry "four situations" monitoring system, unmanned aerial vehicle (UAV) flight defense, intelligent irrigation and fertilization, intelligent greenhouse construction, precision feeding and other agricultural production fields to promote the modernization and precision of agricultural production, minimize agricultural production costs and effectively avoid agricultural business risks [24]. Second, by virtue of digital technology, products are a high-end, green and standardized brands, maximizing exports to the international market and improving the current situation of China's agricultural trade deficit [25]. Third, it is necessary to accelerate the return and concentration of urban related agricultural enterprises and accelerate the intensive and large-scale operation of agriculture by relying on smart agricultural science and technology industrial parks.

2.2.2 Empowering an efficient agricultural operation and management system

Digital technology promotes the precise and optimized allocation of agricultural production factors, complements the short board of the agricultural industry chain, and effectively promotes the establishment of a relationship between the production end and the consumption end [25]. Relying on an agricultural information service (AIS) can boost the brand promotion of agricultural products, integrate agricultural experience, handicrafts, leisure tourism and other elements, and invigorate the interest of all parties in the agricultural industry chain. Building a digital information decision-making platform system is necessary to promote the rapid decision-making response of agricultural operation subjects, improve the accuracy of decision-making, and realize the efficient operation of agricultural operation systems.

2.3 Empowering rural governance

2.3.1 Empowering the digitalization of rural planning

Ecological planting and breeding zones with distinctive regional endowments should be built based on local conditions, and the layout of the village road network should be optimized to eliminate dead-end roads and achieve uniform transportation between urban and rural areas. Relying on a big data platform will help build a new type of livable rural community with an appropriate scale and complete functions, and it will enhance the awareness and provision of social services to rural households. In rural planning and construction, cement irrigation systems should be avoided to prevent the destruction of the original ecological food chain and avoid backtracking and resource waste. 

2.3.2 Enabling public services to be digitized

In basic compulsory education, the construction of rural digital campuses should be vigorously promoted. Digital education infrastructure and cloud platforms for the sharing of educational resources should be built to meet the hardware infrastructure requirements of rural digital compulsory education. In mass education, digital technology should be adopted to build developmental digital infrastructure, such as urban-rural interconnections, digital TV, and digital libraries. The multichannel construction of learning resources is conducive to the realization of educational equality.

By leveraging new digital technologies, an integrated smart medical platform for urban and rural areas should be built to form a comprehensive medical network system integrating expert databases, and patient information and medical records, and to achieve seamless connection between high-quality medical resources in cities and rural areas [26]. A big data platform promotes the interconnection and data sharing between rural medical and health institutions and urban hospitals and realizes the online settlement of medical insurance in different places.

New digital technologies will be used to build digital application platforms for rural life, such as smart rural logistics systems, smart monitoring systems and network interaction systems, to form a complete, closed-loop service system for rural people. With the help of AISs, policies and regulations on agricultural subsidies, the publicity of village affairs, agricultural production and sales markets can be made public and transparent.

2.3.3 Enabling the digitalization of rural governance

The level of digitalization can significantly promote the accurate identification of low-income rural groups, eliminate information asymmetry within rural areas, and improve the accurate identification rate. The layer-by-layer implementation of the top-level system needs to fully rely on blockchain technology and rural information public service platforms. Only in this way, can we improve the transparency of financial support for agriculture and the precision of policy implementation.

The capacity gap and information gap between local governments and rural participants are gradually narrowing [11]. The consciousness of rural subjects to participate in rural governance is gradually enhanced, which fundamentally improves the participation of local people in rural governance. Digital technology has broken the traditional spatial pattern of governance, and rural governance entities have realized cross-regional and cross-temporal deep communication and interaction.

A data cloud service platform provides cloud services for application assistance, medical subsidies and other businesses, and it improves the efficiency of public services. Smart government departments can accurately identify the public demands of rural subjects and realize “proactive plus precise” service-oriented rural governance. They can t carefully consider the hidden dangers existing in various subjects in rural areas in a timely manner and realize the governance mode of “tracking after the event” and “warning in advance”.

---

## [Editor Report · Decision Letter 1]

22 Jun 2022

Utility analysis of digital villages to empower balanced urban-rural development based on the three-stage DEA-Malmquist model

PONE-D-22-02890R1

Dear Dr. Niu,

We’re pleased to inform you that your manuscript has been judged scientifically suitable for publication and will be formally accepted for publication once it meets all outstanding technical requirements.

Kind regards,

Hao Xue

Academic Editor

PLOS ONE

---

## [Editor Report · Acceptance letter]

13 Jul 2022

PONE-D-22-02890R1 

Utility analysis of digital villages to empower balanced urban-rural development based on the three-stage DEA-Malmquist model 

Dear Dr. Niu:

I'm pleased to inform you that your manuscript has been deemed suitable for publication in PLOS ONE. Congratulations! Your manuscript is now with our production department. 

Kind regards, 

on behalf of

Dr. Hao Xue 

Academic Editor

PLOS ONE